# Genistein, a Potential Phytochemical against Breast Cancer Treatment-Insight into the Molecular Mechanisms

Md Sohel [1,2], Partha Biswas [3,4], Md. Al Amin [5], Md. Arju Hossain [5], Habiba Sultana [5], Dipta Dey [6], Suraiya Aktar [7], Arifa Setu [1], Md. Sharif Khan [4], Priyanka Paul [6], Md. Nurul Islam [8], Md. Ataur Rahman [9,10,11], Bonglee Kim [10,11,*] and Abdullah Al Mamun [1,*]

1   Department of Biochemistry and Molecular Biology, Mawlana Bhashani Science and Technology University, Tangail 1902, Bangladesh; mdsohel3921@gmail.com (M.S.); arifashetu10@gmail.com (A.S.)
2   Pratyasha Health Biomedical Research Center, Dhaka 1230, Bangladesh
3   Department of Genetic Engineering and Biotechnology, Faculty of Biological Science and Technology, Jashore University of Science and Technology (JUST), Jashore 7408, Bangladesh; partha_160626@just.edu.bd
4   ABEx Bio-Research Center, East Azampur, Dhaka 1230, Bangladesh; sharifbge42@gmail.com
5   Department of Biotechnology and Genetic Engineering, Faculty of Life Science, Mawlana Bhashani Science and Technology University, Tangail 1902, Bangladesh; alamin.mbstu.bge13@gmail.com (M.A.A.); arju.primer60@gmail.com (M.A.H.); habibasultana725@gmail.com (H.S.)
6   Biochemistry and Molecular Biology Department, Life Science Faculty, Bangabandhu Sheikh Mujibur Rahman Science and Technology University, Gopalganj 8100, Bangladesh; diptadey727@gmail.com (D.D.); paul.bmb011@gmail.com (P.P.)
7   Department of Biochemistry and Molecular Biology, Rajshahi University, Rajshahi 6205, Bangladesh; aktersuraiyaa@gmail.com
8   Department of Pharmacy, Faculty of Life Science, Mawlana Bhashani Science and Technology University, Tangail 1902, Bangladesh; nurul205@gmail.com
9   Global Biotechnology & Biomedical Research Network (GBBRN), Department of Biotechnology and Genetic Engineering, Faculty of Biological Sciences, Islamic University, Kushtia 7003, Bangladesh; ataur1981rahman@hotmail.com
10  Department of Pathology, College of Korean Medicine, Kyung Hee University, Seoul 02447, Korea
11  Korean Medicine-Based Drug Repositioning Cancer Research Center, College of Korean Medicine, Kyung Hee University, Seoul 02447, Korea
*   Correspondence: bongleekim@khu.ac.kr (B.K.); mamunbmb@mbstu.ac.bd (A.A.M.)

**Abstract:** Breast cancer (BC) is one of the most common malignancies in women. Although widespread successful synthetic drugs are available, natural compounds can also be considered as significant anticancer agents for treating BC. Some natural compounds have similar effects as synthetic drugs with fewer side effects on normal cells. Therefore, we aimed to unravel and analyze several molecular mechanisms of genistein (GNT) against BC. GNT is a type of dietary phytoestrogen included in the flavonoid group with a similar structure to estrogen that might provide a strong alternative and complementary medicine to existing chemotherapeutic drugs. Previous research reported that GNT could target the estrogen receptor (ER) human epidermal growth factor receptor-2 (HER2) and several signaling molecules against multiple BC cell lines and sensitize cancer cell lines to this compound when used at an optimal inhibitory concentration. More specifically, GNT mediates the anticancer mechanism through apoptosis induction, arresting the cell cycle, inhibiting angiogenesis and metastasis, mammosphere formation, and targeting and suppressing tumor growth factors. Furthermore, it acts via upregulating tumor suppressor genes and downregulating oncogenes in vitro and animal model studies. In addition, this phytochemical synergistically reverses the resistance mechanism of standard chemotherapeutic drugs, increasing their efficacy against BC. Overall, in this review, we discuss several molecular interactions of GNT with numerous cellular targets in the BC model and show its anticancer activities alone and synergistically. We conclude that GNT can have favorable therapeutic advantages when standard drugs are not available in the pharma markets.

**Keywords:** genistein; breast cancer; molecular pharmacology; anticancer mechanism; synergistic activity

## 1. Introduction

Breast cancer (BC) is considered a major public health problem globally. According to "Cancer statistics, 2020", BC is responsible for around 15% of all cancer-related deaths in females and is the world's third leading cause of mortality among cancer-related deaths [1]. Although there are some common risk factors including aging, sex, gene mutations, family history, and unhealthy lifestyle [2] that can increase the possibility of developing BC, abnormal hormones, namely estrogen, play an effective role in BC progression [3]. Various chemotherapeutic agents are currently available and have been utilized to treat BC for more than half a century [4]; however, a standard cure for the disease still cannot be found in clinical trials. Some existing drugs cause numerous detrimental side effects, including a reduction in blood cells [5], sore throat, hair loss, ulcers, fatigue, nausea, change in taste, appetite loss, constipation, diarrhea, change in skin color, and changes in several hormone levels [6], and some limitations are high costs, low effectiveness, and allergic reactions [7]. Moreover, multidrug-resistant (MDR) tumor formation is the major limitation of conventional treatment, leading to increased cancer-related deaths. Numerous drug molecules, such as anthracyclines (doxorubicin, mitoxantrone, epirubicin), taxanes (docetaxel, paclitaxel), and capecitabine were previously used successfully, but now, cancer patients are becoming resistant to these drugs [8]. However, new plant-based phytochemicals from natural origins may be reliable therapeutic constituents for treating numerous diseases, from infections [9] to cancers [10,11] in humans. Therefore, modern medical science emphasizes better treatments for BC with natural nutritional components [12–14]. Under the phytochemicals, phytoestrogens are a natural dietary component with potent anticancer activity against multiple cancers, most importantly, ovarian, prostate, colorectal, and breast [15–18].

Genistein (GNT) is a soy-based phytoestrogen and is consumed regularly by Asian populations [19]. This phytoestrogen may be one of the leading compounds as its safe and anticancer activities have already been tested in several in vitro and preclinical models. GNT has a structural similarity to 17 β-estradiol, and it binds to estrogen receptor ER-β with higher affinity compared to ER-α [20,21]. Several studies suggested that GNT exerts pleiotropic effects, including inhibiting the cell cycle [22], inducing the cellular apoptosis process [23,24], suppressing metastasis [25] and angiogenesis [26], modulating oxidative stress [27], and mammosphere formation [28] in in vitro BC models. Furthermore, this phytoestrogen exerts several synergistic activities, as it can enhance the efficacy of conventional drugs against BC and reduce chemotherapeutic drug resistance [29]. Moreover, many in vivo [30] and clinical trials [31] also support that GNT can be considered a promising chemopreventive agent for treating different types of BC.

However, besides the cancer-fighting properties, some contradictory results, including promoting malignant cell growths [32] and activating the ATP-binding cassette subfamily C member 1 (ABCC1) protein, makes GNT more difficult to use as an anticancer agent. Therefore, we summarize the available evidence on the chemopreventive and therapeutic potentials of GNT in BC as follows: first, we discuss the molecular pharmacology of GNT in breast tissue; second, we assess molecular mechanisms and synergistic mechanisms to determine the relationship between GNT intake and BC risk; third, we review possible mechanisms of overcoming the resistance of some anticancer drugs.

## 2. An Overview of Genistein

GNT (IUPAC: 5,7-dihydroxy-3-(4-hydroxyphenyl)chromen-4-one) is a phytoestrogen isoflavone that is widely available in soybean, mature seeds, and raw soy-related food (5.6–276 mg/100 g) [32] and legumes (0.2–0.6 mg/100 g) [33]. It possesses lower oral bioavailability, perhaps due to its high solubility in several polar solvents such as acetone, dimethylsulfoxide, and ethanol, and its poor solubility in water [34]. The oral administration of GNT results in high absorption with a $t_{max}$ (transport maximum) of 5–6 h and t1/2 of 8 h [35,36]. GNT is rapidly distributed throughout the body by crossing the placental and blood–brain barriers. GNT is most abundant in the gastrointestinal tract and liver tissue

distribution, consistent with its enterohepatic recycling [37]. GNT is absorbed rapidly and nearly completely in vivo. It showed high permeability in Caco-2 ($3 \times 10^{-5}$ cm/s) and Madin–Darby canine kidney (MDCKII) cells, where passive diffusion is the major transport mechanism, but breast cancer resistance protein (BCRP) may play a role in limiting its intestinal absorption [38–40]. In vivo, GNT undergoes a complex and extensive metabolic process that includes oxidation, reduction, conjugation, glucuronidation, sulfation, and limited CYP reaction [41–46]. Coldham et al. found that GNT has the highest concentrations in the gut (18.5 µg/g), followed by the liver (0.98 µg/g), plasma (0.79 µg/g), and reproductive tissues (uterus, ovary, vagina, and prostate, ranging from 0.12 to 0.28 µg/g) in rats [47]. The excretion of GNT depends on the activity of conjugating enzymes and relies on the efflux transporters' capacity [48]. In vivo, ADME studies revealed that GNT metabolites are excreted via the intestinal, biliary, and renal tracts [49,50]. Although there is limited evidence that consuming large amounts of GNT in the diet causes a deleterious effect in humans, the toxicity of GNT on fertility and fetal development has been extensively studied in recent years. Several studies have demonstrated that therapeutically relevant doses of GNT have a harmful effect on BC differentiation, the estrous cycle, and fertility in rodent models [51,52]. This natural phytochemical can exhibit a wide range of important therapeutic activities, including antioxidant [53], anti-inflammatory [54], antibacterial [55], antiviral [56], antidiabetic [57], and anticancer activities [58]. GNT has proven its ability against various types of human cancers such as lung [59], liver [60], prostate [61], pancreatic [62], skin [63], cervical [64], uterine [65], colon [66], kidney [67], bladder [68], neuroblastoma [69], gastric [70], esophageal [71], pituitary [72], salivary gland [73], testicular [74], ovarian [75], and finally, breast cancer [29].

## 3. Molecular Pharmacology of Genistein in Breast Tissue

GNT is a natural phytochemical belonging to phytoestrogen and it possesses a similar structure to estrogen. Interestingly, it has both mimic and antagonized estrogen effects; simultaneously, it inhibits BC cell proliferation [76]. Estrogen receptor-mediated growth regulation of BC cells by GNT may be concentration-dependent. T.T.Y. Wang et al. summarized that GNT stimulated growth at lower concentrations ($10^{-8} \sim 10^{-6}$ M), but inhibited cancer cell growth at higher concentrations ($>10^{-5}$ M) [77]. There are two types of estrogen receptors [78]. GNT has a structural similarity to both ER-$\alpha$ and ER-$\beta$ receptors but binds with ER-$\beta$ with higher affinity compared to ER-$\alpha$ [20,21]. In the case of ER-$\alpha$, GNT acts as an antagonist. Thus, GNT-mediated anticancer activity is involved by suppressing the expression and activity of ER-$\alpha$. E.J. Choi summarized that GNT regulates cell proliferation with apoptosis via the ER-$\alpha$-dependent pathway in MCF-7 BC cells through the underlying mechanism of downregulating cyclin D1 and upregulating the Bcl-2/Bax ratio (B cell lymphoma 2/BCL associated X) at the dose of 50 µM [79]. On the other hand, in ER-$\beta$, GNT increases receptor activities as a type of agonist. Therefore, ER-$\beta$-dependent anticancer activity of GNT is mediated by activating the receptor and potentiating chemotherapeutic efficacy to treat cancer [80]. H. Jiang stated that GNT mediated anticancer activities through ER-$\beta$1 receptors in MDA-MB-231, MCF-7 cells, and BALB/c mice by inhibiting cell proliferation through arresting cells in the G2/M and G0/G1 phases, which led to cell cycle blockade at the dose of $10^{-6}$–$10^{-4}$ mol/L [81].

It has also been found that GNT can bind with the estrogen-responsive G protein-coupled receptor-30 (GPR-30) or G protein-coupled estrogen receptor-1 (GPER-1) [82] and inhibit cell proliferation [83]. Kim GY et al. summarized that GNT suppresses GPR-30 activation in breast cancer gene 1(BRCA-1)-mutated BC cells, resulting in G2/M phase arrest mediated by suppressing Akt phosphorylation [84]. Human epidermal growth factor receptor 2 (HER-2) is an important biomarker in BC and overexpressed in around 20–30% of BC types [85]. Thus, regulating HER-2 is a significant factor in BC treatment. Sakla et al. summarized that GNT inhibited proto-oncogenes of HER-2 and subsequently followed the HER-2 protein expression, phosphorylation, and promoter activity through an ER-independent mechanism in BC cells, aiming to delay tumor onset in transgenic mice [86].

Furthermore, GNT can inhibit protein tyrosine kinase (PTK), hypothesized to be responsible for the lower rate of BC observed in Asian women consuming soy. Akiyama et al. reported that GNT scarcely inhibited the enzyme activities of threonine- and serine-specific protein kinases such as cAMP-dependent protein kinase, $Ca^{2+}$/phospholipid-dependent enzyme protein kinase C, and phosphorylase kinase, and these mechanisms are mediated through phosphorylation of the EGF receptor [87].

## 4. Cell-Specific Molecular Mechanisms of Genistein-Mediated Anti-Breast Cancer Activity In Vitro

Cancerous cell lines derived from humans are critical models for in vitro cancer research to determine the therapeutic advantage of anticancer agents [88]. Anticancer activity of phytochemicals is cell-specific, where one phytochemical is effective in one or more cell lines, and this may be the difference in the cell components system. Cell line-specific anticancer activity of GNT is summarized in Figure 1.

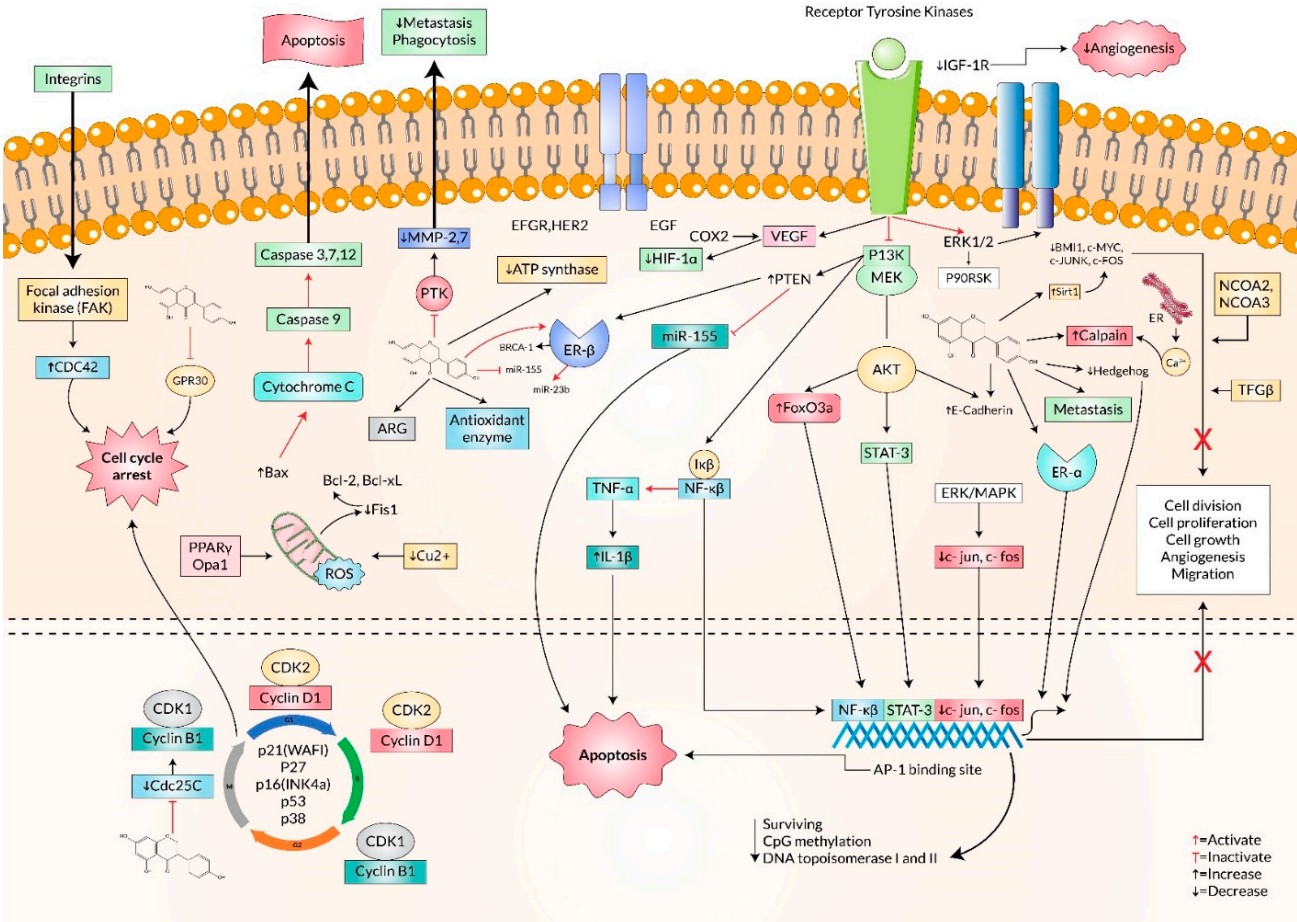

**Figure 1.** Graphical overview of the anticancer mechanisms of genistein. GNT induces apoptosis through a mitochondrial-mediated classical caspase-dependent pathway with modulating Bcl-2 family proteins. It induces cell cycle arrest by modulating the cycle regulatory proteins. It inactivates signaling pathways, namely PI3K/AKT and MAPK (ERK1/2) pathways. GNT also modulates several miRNA expressions and suppresses cell migration, invasion, and angiogenesis, and regulates epigenetic control.

### 4.1. The Effects of Genistein on MCF-7 BC Cells

According to Prietsch et al., GNT (0.01–100 μM) promoted apoptosis via mediating the autophagy-dependent mechanism and increasing the ratio of Bax/Bcl-2 and inhibiting the oxidative stress of cancer progression through changing the expression of antioxidant enzymes [89]. Liu et al. summarized that GNT (5–20 μM) induced apoptosis through the mitochondrial-dependent pathway by decreasing the Bcl-2/Bax ratio and increasing tumor suppressor gene p73 expression and ATM phosphorylation with G2/M phase arrest permanently [90]. Similarly, GNT (50–200 μM) halted cellular growth and induced apoptosis by following the downregulation of Bcl-2 protein, upregulation of Bax, and decreasing cyclin D1 expression in the MCF-7 BC cell line [79]. At a low concentration, GNT (1 μM) stimulates cell proliferation, but a higher concentration (25 μM) induces apoptosis pathways by upregulating the CDKN1A and p53 responsive genes and downregulating CCNG1 GADD45A, NF-κB, Bcl-2, TNFR, ESR1, NCOA2, and NCOA3 [91]. Another study investigated that GNT (50 μM) induced apoptosis by upregulating poly-(ADP-ribose)-polymerase and p53, and downregulating the Bcl-2/Bax protein ratio [92]. An in vitro study by Lemos investigated that GNT (10 μM) induced apoptosis by breaking the plasma membrane, nuclear membrane, and upregulating pS2 expression [93]. A later study reported that GNT (100 μM) mediated programmed cell death and suppressed cell growth by upregulating caspase 7, apoptosis signaling kinase-1, ADP ribose, and p38-dependent mitogen protein kinase [94]. Inhibition of metastasis and angiogenesis processes is a common mechanism in BC treatment. In vitro study demonstrated that GNT (3.125–12.5 μM) decreased tumorigenic processes by increasing GSTP1 and RARβ2 gene expression and activity [95]. Shon et al. concluded that GNT suppressed angiogenesis by downregulating COX, TPA, and EROD proteins [96], while at 1–10 μg/mL, it inhibited angiogenesis by decreasing tyrosine kinase and ribosomal S6 kinases [97]. In an in vitro study, GNT lowered cell proliferation via mitochondrial-dependent pathways by reducing Fis1 (mitochondrial fission) and Opa1 (mitochondrial fusion) mRNA expression [98] at 10 nm–10 μM, while 4–10 mol/L of GNT inhibited cell proliferation by downregulating cyclin D1 and arresting the cell cycle in the G0/G1 phase, resulting in the blockage of cell survival, according to H. Jiang et al. [81].

Chen et al. reported that GNT (5–100 μM) inhibited the proliferation of cells by inducing apoptosis through IGF-1R-PI3 K/Akt-mediated pathway inactivation and upregulating the Bax/Bcl-2 ratio [99]. Furthermore, it has been shown that GNT (5–30 μM) inhibited BC cell growth, proliferation, and promoted apoptosis by following the downregulation of the Hedgehog–Gli1 signaling pathway and decreasing the mRNA level of Smo and Gli1 [100]. Marik et al. also found similar results, that GNT at a low concentration (0.1 μM) stimulates cancer progression, but GNT (20 μM) at a high concentration inhibits cell proliferation by downregulating mRNA expression of ER-α protein and arresting the cell cycle at the G2/M phase [76]. Furthermore, Chinni et al. reported that GNT (100 μM) inhibits cell proliferation by downregulating Akt-mediated signaling pathways, decreasing telomere length, and overexpression of cyclin-dependent kinase inhibitor p21WAF1 [101]. An early study demonstrated that GNT (50 μM) inhibited tumor growth with apoptosis inductions by increasing $Ca^{2+}$-dependent pro-apoptotic protease, mμ-calpain, and caspase-12 [102]. On the other hand, Liao et al. showed that GNT (100 μM) inhibited cell growth alongside decreasing paclitaxel-induced tubulin polymerization, Bcl-2, cyclin B1, and CDK2 kinase, leading to cell cycle arrest at the G2/M phase [103]. Chen et al. showed that GNT (50–100 μM) suppressed cell division through uplifting heat shock protein (HSP) activity and reducing SRF mRNA, RAG-1, and DOC 2 expression [104]. GNT (40 nm–2 μM) inhibits mammosphere formation in BC stem cells by suppressing PI3K/Akt signaling through upregulating the PTEN expression [28]. A similar result found by Y. Liu et al. confirmed that GNT (40 nm–2 μM) inhibited mammosphere formation and induced stem cell differentiation by activating PI3K/Akt and MEK/ERK signaling in a paracrine manner, increasing E-cadherin mRNA expression by reducing the ratio of CD44+/CD24-/ESA in MCF-7 BC cells [105]. GNT (1 μM) induces an anticancer effect through upregulating

pro-inflammatory genes, i.e., pS2 and COX2, and downregulating anti-inflammatory gene expression, i.e., TFGβ and PPARγ in MCF-7 BC cells [106]. Furthermore, Kazi et al. reported that GNT (50–200 μM) halts cancer progression by upregulating IκB-α and p27 (Kip1) levels, and downregulating proteasomal chymotrypsin-like activity and CDKs [107]. Epigenetics regulation by GNT (60–100 μM) is mediated by diminishing DNA methylation levels, DNMT1 expression, and DNA methyltransferase enzyme activity. However, this reduction in DNA methylation occurs in the promoter region of multiple tumor suppressor genes (TSGs) such as adenomatous polyposis coil (APC), ataxia telangiectasia mutated (ATM), phosphatase and tensin homolog (PTEN), and mammary serpin peptidase inhibitor (SERPINB5) [108].

*4.2. The Effects of Genistein on MDA-MB-231 BC Cells*

Recently, an experiment conducted by Liu et al. GNT (5–20 μM) induced apoptosis through the mitochondrial-dependent pathway by reducing the Bcl-2/Bax ratio and inhibiting cell growth and increasing the expression of p73, leading to the activation of G2/M phase arrest and the ATM/Cdc25C/Chk2/Cdc2 checkpoint pathway [90]. GNT prompted the apoptotic pathway and directly inhibited the growth of cells through the prevention of NF-κB signaling by the Notch-1 pathway and by downregulating cyclin B1 and Bcl-2 expression, resulting in the arrest of the cell cycle at the G2/M phase at 5–20 μM [109], while at 5–50 μM, this phytochemical induced apoptosis by targeting the endogenous copper ion, reducing Cu(II) to Cu(I) through the production of reactive oxygen species (ROS) [110]. Before that, an in vitro study by Dampier et al. reported that GNT (10 μM) induced apoptosis and inhibited cell proliferation and cell cycle arrest at the G2 phase, degrading proto-oncogene *c-Fos* and prohibiting protein-1 (AP-1), and also ERK activity [111]. Another study by Yang et al. demonstrated that GNT (50 μM) exerted apoptosis by upregulating poly-(ADP-ribose)-polymerase, activating p53, and downregulating Bcl-2/Bax protein [92].

In the case of angiogenesis, Mukund et al. explained that GNT (100 μM) reduced angiogenesis by blocking the transactivation of downstream HIF-1α effectors, e.g., VEGF, leading to the reduction in hypoxia-inducible factor-1α expression in MDA-MB-231 BC cells [26]. Furthermore, 1–10 μg/mL of GNT suppressed angiogenesis and cell mutation by decreasing tyrosine kinase, ribosomal S6 kinases, and DNA topoisomerases I and II [97], while at a 50 μM concentration, it decreased angiogenesis and inhibited cell division through the underlying mechanism of downregulating COX, topoisomerase II enzyme TPA, and EROD protein activity [96]. Followed by angiogenesis, GNT (15–30 μM) [112] and (5–20 μM) [24] obstructed cancer cell migration and invasion, respectively, by lowering levels of CDKs, tyrosine kinase, and paracrine stimulation and decreasing MEK5, ERK5, phospho-ERK5, NF-κB/p65, and Bcl-2/Bax.

Another study conducted by Kousidou et al. reported that GNT (35–100 μM) progresses slowdown invasiveness by decreasing MMP gene expression, PTK activity, and glucose uptake rate, leading to phagocytosis of cancerous cells [113]. Apart from this, it reduces cell viability by decreasing the DNA methyltransferase activity and DNMT1 expression and affecting the expression of TSGs, i.e., APC, ATM, PTEN, and SERPINB5 at 60–100 μM of GNT [108]. Another recent study by Pons et al. summarized that GNT (1 μM) causes a considerable decrease in cell viability through the mitogen-dependent protein kinase pathway and by promoting apoptosis mechanisms [106].

In MDA-MB-231 BC cells, cell growth control is a significant target for GNT. Gong et al. stated that GNT (5–50 μM) inhibited cell growth by partly inducing apoptosis via downregulation of the Akt and NF-κB cascade pathways [114]. In another in vitro analysis, the cell growth inhibitory activity was evidenced by GNT (2.5–400 μM) through the upregulation of two crucial TSGs, p21WAF1 (p21) and p16INK4a (p16), and the downregulation of two tumor-promoting genes, c-MYC and BMI1, ultimately inhibiting cancer progression [115]. Y. Fang et al. concluded that GNT (40 μM) inhibited cellular growth by following the activation of DNA-dependent damage response and the ATR signaling

pathway and activating the BRCA-1 complex, inhibiting the cohesion complex, and increasing phosphatide, which is distributed among CDK1, CDK2, and CDK3 [116]. Recently, it was established that GNT (1000 ppm) suppressed tumor growth by cell cycle regulation via maintaining the expression level of the cyclin D1 protein, leading to G0/G1 phase arrest, which causes cell cycle blockage [81]. Subsequently, Rajah et al. summarized that GNT (10–100 μM) inhibited tumor growth by downregulating MEK5, pERK5, and NF-κB proteins [117]. In the case of cell proliferation, a low dose of GNT (10 μM) slightly inhibited cell proliferation by reducing the P-STAT3/STAT-5 ratio [98]. In comparison, at a double dose, i.e., 20–40 μM, it significantly prevented cell proliferation by inducing apoptosis and suppressing Skp2 expression by upregulating the tumor suppressor genes, i.e., p21 and p27, resulting in G2/M phase arrest [118]. Li et al. investigated that GNT (5–20 μM) inhibited cell differentiation with cell cycle arrest at the G2/M phase by decreasing CDK1, cyclin B1, Cdc25C, c-Jun, and c-Fos levels [22]. GNT can also play a role in MDA-MB-231 by inhibiting mammosphere formation. A lower dose of GNT (2 μM) prevents mammosphere formation through PI3K/Akt signaling by increasing the PTEN expression [28], while at a higher dose, GNT (40 nm–2 μM) prevents the formation of mammosphere cells and promotes differentiation through the PI3K/Akt and MEK/ERK signaling pathway by reducing the CD44+/CD24-/ESA ratio and increasing E-cadherin mRNA expression [105]. Finally, GNT (50 μM) impedes primary tumor formation by downregulating chelator neocuproine and Bcl-2/Bax and by upregulating the caspase-3 pathway [110].

*4.3. The Effects of Genistein on T-47D Breast Cancer Cells*

Mukund et al. summarize that GNT (50 μM) lowered angiogenesis by preventing the transactivation of downstream HIF-1α effectors such as VEGF, reducing the expression of hypoxia-inducible factor-1α in the T-47D BC cell line [26]. Cell proliferation efficacy was evident by GNT (10 nm) with apoptosis induction through the mitochondrial-dependent pathway via upregulating the cyt-C and oxidase activity, and downregulating the ATP synthase/cytochrome c oxidase ratio [98]. GNT at 1 nm–100 μM inhibits cell proliferation through ERK1/2-mediated signaling by the downregulation of phosphorylated p90RSK [119], while 10 μM of GNT induces apoptosis and inhibits cell proliferation through degrading proto-oncogene c-Fos levels and prohibiting protein 1 (AP-1) and ERK expression [111]. Another in vitro study by Rajah revealed that GNT (10–100 μM) inhibits cell proliferation and tumor growth by downregulating MEK5, pERK5, and NF-κB proteins [117]. Additionally, a high GNT (20 M) concentration inhibits cell proliferation by reducing ER-messenger RNA transcription and arresting the cell cycle at the G2/M phase [76]. According to Sotoca et al., GNT (500 nm) inhibited cell growth and induced apoptosis by activating cytoskeleton restructuring that results in interaction among integrins, focalized adhesion kinase, and CDC42 that leads to cell cycle arrest in the T-47D BC cell line [120], while according to Pons et al., GNT (1 μM) caused a significant decrease in cell viability by increasing Sirt1, TGFβ, and PRARγ and decreasing IL-1β expression in T-47D BC cells [106].

*4.4. The Effects of Genistein on HCC1395 Breast Cancer Cells*

Lee et al. demonstrated that GNT (1–200 μM) inhibited HCC1395 cell invasion and metastasis through the upregulation of TFPI-2, ATF3, DNMT1, and MTCBP-1 gene expression and the downregulation of MMP-2, MMP-7, CXCL12 genes, leading to cell cycle arrest at the G2/M phase, therefore reducing cell viability [25].

*4.5. The Effects of Genistein on HCC38 Breast Cancer Cells*

Donovan stated that GNT (4–10 ppm) inhibited cell growth by increasing the BRCA1 protein level and reducing CpG methylation, consequently decreasing the aryl hydrocarbon receptor (AhR) binding at BRCA1 in the HCC38 cell line [121].

### 4.6. The Effects of Genistein on Hs578t Breast Cancer Cells

According to Parra et al., GNT (1–50 μM) inhibits cell viability and induces apoptosis through the downregulation of mir-155, resulting in the upregulation of casein kinase, FOXO3a, p27, and PTEN expression, and the reduction of β-catenin in the Hs578t cell line [122].

### 4.7. The Effects of Genistein on DD-762 and Sm-MTC Breast Cancer Cells

Nakagawa et al. appraised that GNT (7–274.2 μM) inhibited cell proliferation by upregulating caspase-3 protein activity in DD-762 and Sm-MTC BC cell lines [123].

### 4.8. The Effects of Genistein on BT-474 Breast Cancer Cells

GNT at a low concentration (1 μM) could promote cancer but at a high concentration (50 μM), it inhibits cell division by downregulating tyrosine kinase, HER2 activation, and the MAPK pathway [86]. GNT (3.125–25 M) inhibits cell replication and arrests the cell cycle in the G2/M phase, and inhibits the expression of EGFR, HER2, and ER-alpha [124].

### 4.9. The Effects of Genistein on BT20 Breast Cancer Cells

Cappelletti et al. revealed that GNT (15–30 μM) inhibits metastasis by lowering levels of CDKs, tyrosine kinase, DNA topoisomerase II, and paracrine stimulation in the BT20 cell line [112].

### 4.10. The Effects of Genistein on 21PT Breast Cancer Cells

Marik et al. demonstrated that GNT at a 0.1 M concentration stimulated cancer progression, while 20 M of GNT inhibited cell proliferation by decreasing ER-messenger RNA expression and arresting the cell cycle at the G2/M phase in the 21PT cell line [76].

### 4.11. The Effects of Genistein on 184-B5/HER Breast Cancer Cells

Katdare et al. showed that GNT (2.5–10 μM) impeded the cell cycle and induced apoptosis by increasing the P16INK4a gene and decreasing HER-2/neu and tyrosine kinase [125].

### 4.12. The Effects of Genistein on MCF-10A, MCF-ANeoT, and MCF-T63B Breast Cancer Cells

An early study showed that GNT (1–10 μg/mL) obstructed angiogenesis and cell mutation by decreasing the expression of ribosomal S6 kinases and tyrosine kinase [97]. An overview of GNT's anticancer activities is given in Table 1.

**Table 1.** Tabular representation of in vitro anti-breast cancer activity of genistein.

| Target | Pharmacological Interaction | Type of Study (In Vitro or In Vivo) | Dose | Molecular Mechanism | Molecular Target | Ref. |
|---|---|---|---|---|---|---|
| | | In vitro (MCF-7) | 5–100 μM | ↑Apoptosis ↓Cell proliferation | ↓IGF-1R-PI3 K/Akt pathway ↓Bcl-2/Bax level | [99] |
| ER-α | Antagonist | In vitro (MDA-MB-231 and T-47D) | 50–100 μM | ↓Angiogenesis | ↓VEGF ↓HIF-1α expression | [26] |
| ER-α ER-β | Agonist Antagonist | In vitro (MCF-7, T47D, and MDA-MB-231) | 0.01–10 μM | ↓Cell proliferation ↓Mitochondrial activity | ↓Opa1, Fis1 ↑Cytochrome c oxidase ↓ATP synthase/cytochrome c ratio ↓P-STAT3/STAT-3 ratio | [98] |
| ER-α ER-β | Not mentioned | In vitro (MDA-MB-231) | 5–20 μM | ↑Arrest G2/M phase ↑Apoptosis ↑Cell cycle arrest | ↓NF-κB, Notch-1 pathway ↓Cyclin B1, ↓Bcl-2 and Bcl-xL | [109] |
| ER-α ER-β | Not mentioned | In vitro (MDA-MB-231) | 2.5–400 μM /250 mg/kg | ↓Cell growth | ↑p21WAF1 (p21), p16INK4a (p16) ↓BMI1, c-MYC | [115] |
| ER-α ER-β | Agonist Antagonist | In vitro (MCF-7 and MDA-MB-231) | 5–20 μM | ↑Arrest G2/M ↑Apoptosis ↓DNA damage | ↑ATM, Bax, P73 ↓Bcl-2, ↓Bcl-2/Bax rate, mutant P53 | [90] |

**Table 1.** *Cont.*

| Target | Pharmacological Interaction | Type of Study (In Vitro or In Vivo) | Dose | Molecular Mechanism | Molecular Target | Ref. |
|---|---|---|---|---|---|---|
| ER-α<br>ER-β | Agonist<br>Antagonist | In vitro<br>(MCF-7, MDA-MB-435, and Hs578t) | 1–50 µM | ↓Cell viability<br>↑Apoptosis | ↓miR-155<br>↓β-catenin<br>↑Casein kinase,<br>↑FOXO3a, p27, PTEN, CK1α<br>↑Phosphopeptide | [122] |
| ER-α<br>ER-β | Not mentioned | In vitro<br>(MDA-MB-231) | 40 µM | ↓Cell growth | ↓Cohesin complex<br>↑DNA damage response pathway<br>↑BRCA1 | [116] |
| ER-α<br>ER-β | Agonist<br>Antagonist | In vitro<br>(MCF-7 and MDA-MB-231) | 40 nm–2 µM | ↓Mammosphere formation | ↑ATR signaling pathway<br>↑PTEN expression<br>↓PI3K/Akt signaling | [28] |
| ER-α<br>ER-β | Agonist<br>Antagonist | In vitro<br>(MCF-7, T47D, and MDA-MB-231) | 1 µM | ↓Cell viability<br>↓Cell proliferation | ↑ROS, pS2, Sirt1, COX2<br>↓IL-1β, TFGβ, PPARγ | [106] |
| ER-α<br>ER-β | Agonist<br>Antagonist | In vitro<br>(T47D) | 500 nm | ↓Cell growth<br>↑Apoptosis | ↑Cytoskeleton remodeling<br>↑Integrins, focal adhesion kinase, CDC42<br>↑Arrest cell cycle | [120] |
| ER-α | Antagonist | In vitro<br>(T47D) | 1–100 µM | ↓Cell proliferation | ↓ERK1/2, p90RSK | [119] |
| ER-α | Antagonist | In vitro<br>(MCF7, UACC3199, and HCC38) | 4–10 ppm | ↓Cell growth | ↑Activate BARCA-1<br>↓CpG methylation<br>↓AHR activity | [121] |
| ER-β | Agonist | In vitro<br>(MCF-7 and MDA-MB-231) | $10^{-6}$ mol/L–$10^{-4}$ mol/L/<br>100–1000 ppm | ↓Cell proliferation | ↓Cyclin D1<br>↑Arrest G0/G1 phase | [81] |
| ER-α<br>ER-β | Not mentioned | In vitro<br>(MCF-7) | 5–30 µM/<br>20–50 mg/kg | ↓Cell growth<br>and proliferation | ↓Hedgehog–Gli1<br>signaling pathway | [100] |
| ER-α<br>ER-β | Not mentioned | In vitro<br>(MDA-MB-231, and MDA-MB-468) | 5–50 µM | ↑Apoptosis | ↓Endogenous copper ion<br>↑Generation of reactive oxygen species (ROS)<br>↑BRCA-1 | [110] |
| ER-α | Antagonist | In vitro<br>(MCF-7 and UACC-3199) | 0.5–20 µM | ↓Cell growth | ↓CpG methylation<br>↓Cyclin D1<br>↓DNMT-1<br>↑AhR<br>↑CYP1A1 | [126] |
| ER-α | Antagonist | In vitro<br>(HCC1395) | 1–200 µM | ↓Cell viability<br>↓Invasion<br>↓Metastasis | ↑TFPI-2, ATF3, DNMT1, MTCBP-1<br>↓MMP-2<br>↓MMP-7, CXCL12<br>↑Arrest G2/M phase | [25] |
| ER-β | Agonist | In vitro (SUM1315MO2) | 1–100 µM | ↓Cell proliferation | ↑ER-β expression<br>↑Restore BRCA1 function | [127] |
| ER-α | Antagonist | In vitro<br>(MCF-7) | 50–200 µM | ↓Cell growth<br>↑Apoptosis | ↓Bcl-2<br>↑Bax<br>↓Cyclin D1<br>↓Bcl-2/Bax ratio | [79] |
| ER-α | Agonist<br>and<br>Antagonist | In vitro<br>MCF-7 | 1–25 µM | ↑Apoptosis<br>↓Cell proliferation | ↑CDKN1A, TNF-α p53 responsive gene<br>↓CCNG1 and GADD45A<br>↓BCL-2, BCL-3, and NF-kappa B and TNFR<br>↓NCOA2 and NCOA3 | [91] |
| ER-α | Antagonist | In vitro<br>MDA-MB-435 | 750 µg/g | ↑Apoptosis<br>↓Metastasis | ↓Tyrosine phosphorylation<br>↑Matrix degrading enzymes | [128] |
| ER-α | Antagonist | In vitro<br>MCF-7, ZR-75.1, T47-D, MDA-MB 468, MDA-MB 231, and HBL | 1–10 µM | ↑Apoptosis<br>↓Cell proliferation | ↓c-Fos levels, protein-1 (AP-1) activity, ERK signal<br>↑Arrest at G2 phase | [111] |
| ER-α<br>ER-β | Not mentioned | In vitro<br>(MCF-7 and MDA-MB-231) | 60–100 µM | ↓Cell viability | ↓mRNA expression of DNMT1<br>↓DNA methylation in tumor suppressor genes | [108] |
| ER-α<br>ER-β | Not mentioned | In vitro<br>(MCF-7) | 0.01–100 µM | ↑Apoptosis | ↑Bax/Bcl-2 ratio | [89] |
| ER-α<br>ER-β | Antagonist | In vitro<br>MCF-7<br>MDA-MB-231 | 50 µM | ↑Apoptosis<br>↓Cell division<br>↓Angiogenesis | ↓EROD, TPA<br>↓Cyclooxygenase<br>↓Tyrosine kinase | [96] |
| ER-α<br>ER-β | Antagonist | In vitro<br>MDA-MB-468 | 25–100 µM | ↓Cell cycle kinetics<br>↑Apoptosis<br>↓Cell proliferation | ↑Arrest at G2/M phase<br>↑Nuclear membrane breakdown during G2/M transition<br>↓DNA synthesis | [129] |
| ER-β | Antagonist | In vitro<br>MCF-7 | 0.001–10 µM | ↑Cell apoptosis | ↑Plasma membrane breakdown<br>↑Nuclear membrane breakdown<br>↑pS2 expression | [93] |
| ER-β | Antagonist | In vitro<br>MCF-7 and MDA-MB-231 | 10–100 µM | ↑Apoptosis<br>↓Cell division | ↓PTK, Akt, FAK, ErbB-2, and Bcl-2<br>↓Topoisomerase II, tyrosine kinase<br>↓Osteoclast activity | [87, 113, 130–137] |
| ER-α<br>ER-β | Not mentioned | In vitro<br>(MDA-MB-231 and SKBR3) | 20–40 µM | ↓Cell proliferation<br>↑Apoptosis<br>↓Metastasis | ↓Skp2 expression<br>↑Arrest G2/M phase<br>↑p21, p27 | [118] |
| ER-α<br>ER-β | Not mentioned | In vitro<br>(MCF-7) | 75–200 µM | ↓Cell growth | ↑miR-23b<br>↑Target PAK2 gene<br>↑Anti-growth signals protein | [138] |
| ER-α<br>ER-β | Antagonist | In vitro<br>MDA-MB-231 | 0.5–15 µM | ↓Cell cycle kinetics<br>↑Apoptosis | ↑Connexin phosphorylation blocks the homeostatic regulators | [139] |

**Table 1.** *Cont.*

| Target | Pharmacological Interaction | Type of Study (In Vitro or In Vivo) | Dose | Molecular Mechanism | Molecular Target | Ref. |
|---|---|---|---|---|---|---|
| ER-α | Antagonist | In vitro MDA-MB-231, BT20, T47D, and ZR75.1 | 15–30 μM | ↑Cell apoptosis ↓Migration | ↑Arrest at G2/M phase ↓Tyrosine kinase ↓Paracrine stimulation | [112] |
| ER-α ER-β | Agonist and Antagonist | In vitro MDA-MB-231 and T47D | 10–100 μM | ↓Decrease cell proliferation ↓Tumor growth | ↓MEK5, pERK5, NF-κB proteins | [117] |
| ER-α ER-β | Antagonist | In vitro AS-4, NEO, and BG-1 | 25–150 μM | ↑Inhibit cell proliferation ↑Induce apoptosis | ↓Cytotoxic effect in AS4, tyrosine kinase ↑Ubiquitin E3 ligase | [140] |
| ER-α | Antagonist | In vitro MDA-MB-231 | 5–20 μM | ↑Trigger apoptosis ↓Invasion | ↓MEK5, ERK5, NF-κB/p65 ↓Bcl-2/Bax | [24] |
| ER-α | Antagonist | In vitro MDA-MB-231 | 5–20 μM | ↑Cell depletion ↓Differentiation | ↑Trigger G2/M cell cycle arrest ↓CDK1, cyclinB1, and Cdc25C ↓c-Jun and c-Fos | [22] |
| ER-α | Antagonist | In vitro MCF-7 | 50 μM | ↓Tumor growth ↑Induce apoptosis | ↑$Ca^{2+}$-dependent proapoptotic proteases ↑mμ-calpain and caspase-12 | [102] |
| ER-β | Agonist and Antagonist | In vitro MDA-MB-231 | 20–80 μM and 750 μg/g | ↓Cell growth ↓Tumor formation | ↑Cell cycle blocked at G2/M | [141] |
| ER-α | Antagonist | In vitro MCF-7 and MDA-MB-231 | 100 μM | ↑Induce apoptosis ↓Cell proliferation | ↑Arrest at G2/M phase ↓Paclitaxel-induced tubulin Polymerization ↓Bcl-2 phosphorylation ↓Cyclin B1 and CDC2 kinase ↓HER2 and ER-alpha | [103] |
| ER-α | Antagonist | In vitro BT-474 | 3.125–25 μM | ↑Induce apoptosis ↓Duplication | ↑Cell cycle arrest at S and G2/M ↓Expression of surviving, EGFR ↓Protein tyrosine kinase pathway | [124] |
| ER-α ER-β | Antagonist | In vitro MDA-MB-231, MCF-7, and MCF-12A | 35–100 μM | ↓Cell invasiveness ↓Cell cycle ↑Phagocytosis | ↑Arrest at G2/M phase ↓MMP genes ↓Glucose uptake | [113] |
| ER-α | Antagonist | In vitro MCF-7 | 100 μM | ↑Induce apoptosis ↓Cell Growth | ↑Caspase 7 and poly (ADP ribose) polymerase ↑Apoptosis signaling kinase 1 ↑p38 MPK | [94] |
| ER-α | Antagonist | In vitro MDA-MB-231 | 5–50 μM | ↑Induce apoptosis ↓Cell cycle | ↑Apoptosis-related genes ↓Akt, NF-κB signal ↓EGF | [114] |
| ER-α | Antagonist | In vitro MCF-7 | 50–200 μM | ↓Cell invasion ↑Apoptosis ↓Cancer progression | ↓Proteasomal chymotrypsin-like activity ↓CDKs inhibit by p27Kip1 ↑IκB-α level ↓SRF, RAG-1, DOC 2 | [107] |
| ER-α | Antagonist | In vitro MCF-7 | 50–100 μM | ↓Cell division ↓Cancer progression | ↑Arrest at G2/M phase ↑Heat shock protein 105 (HSP) mRNA | [104] |
| ER-α | Agonist Antagonist | In vitro MCF-7 and MDA-MB-468 | 3.125–12.5 μM | ↓Primary tumor ↓Metastasis | ↑GSTP1 gene ↑RARβ2 gene | [95] |
| ER-α | Antagonist | In vitro MCF-7 | 1–100 μM | ↑Apoptosis, cell cycle arrest | ↓CDKs, Akt activity ↓Telomere length | [101] |
| ER-α | Agonist Antagonist | In vitro MCF-7, 21PT, and T47D | 0.1–20 μM | ↓Cell proliferation ↓Cancer progression | ↓ER-α messenger RNA expression ↑Cell cycle arrest at the G2-M phase | [76] |
| | Antagonist | In vitro MDA-MB-231 and MDAMB-468 | 50 μM | ↓Primary tumor ↑Apoptosis | ↓Chelator neocuproine ↓Bcl-2 ↑Bax and caspase-3 ↑P16INK4a gene | [110] |
| ER-α | Antagonist | In vitro 184-B5/HER | 2.5–10 μM | ↑Cellular apoptosis ↓Cell cycle regulators | ↓HER-2/neu ↓Tyrosine kinase ↑Arrest at S + G2/M phase | [125] |

## 5. Clinical Trials

Human clinical trials have confirmed the in vitro research findings. In some cases, when consumed at a consistent dose, pure GNT had no estrogenic effect on breast tissue [31,142], although in other cases, dietary soy supplementation had pro-estrogenic effects on breast tissue [143–145]. Several secondary endpoints were evaluated in a recently published clinical study to determine whether purified GNT affects endometrial thickness, vaginal cytology, and breast density (Table 2 [31,95,96]. Following the implementation of safety measures, it was possible to identify the potential estrogenic effects of 54 mg/day of purified GNT as indicators of BC risk in the research participants. Indeed, while the placebo group maintained a constant endometrial thickness, the GNT group demonstrated a time-dependent reduction that reached statistical significance during the 36-month follow-up (approximately 12% reduction, $p < 0.01$). Moreover, levels of gene expression of BRCA-1 and 2 breast tumor suppressor genes [146,147] have been preserved for three years in the GNT-administered group, while levels of both BRCA-1 and 2 have decreased in the placebo

group (Table 2) [31,142]. GNT also significantly reduced sister chromatid exchanges, implying that it may prevent genotoxicity and subsequent mutagenesis (Table 2) [142]. In this regard, based on the use of GNT in BC, two clinical trials—a phase II study entitled "Gemcitabine Hydrochloride and GNT in Treating Women with Stage IV BC" (NCT00244933) and a phase I study entitled "GNT in Preventing Breast or Endometrial Cancer in Healthy Postmenopausal Women" (NCT00099008)—have been completed, but the results are not yet published. The effects of GNT on human clinical studies against cancer are summarized in Table 2.

**Table 2.** Effects of genistein on markers of cancer risk observed from human clinical studies (postmenopausal women).

| Subjects (*n*) | Number/Intervention (mg/day) | Study Length (Months) | Results | Ref |
|---|---|---|---|---|
| 57 | 54g/day GNT (*n* = 30) or placebo (*n* = 27) | 12 | Protect genomic damage | [142] |
| 389 | Placebo (*n* = 191; 1000 mg calcium and 800 mg vitamin D) or GNT (*n* = 198; 54 mg GNT, 1000 mg calcium, and 800 mg vitamin D) | 24 | ↑BMD at the lumbar spine and the femoral neck ↑B-ALP and IGF-1 | [148] |
| 220 | Placebo (*n* = 111) or isoflavone (*n* = 109; ~50 mg isoflavones) | 24 | ↓Breast area ↓Breast density; neither was significant when compared to placebo | [149] |
| 34 | Placebo (*n* = 17) or soy (*n* = 17; 100 mg isoflavone, ~76 mg aglycones) | 12 | ↓Breast area | [150,151] |
| 84 | Placebo (*n* = 23) or soy (*n* = 28; 60 mg, ~45 mg isoflavones) | 14 days | ↓Nipple aspirate levels of apolipo-protein D ↑pS2 level unchanged breast density, BRCA1 and BRCA2 | [143,144] |
| 389 | 54 mg of GNT aglycone daily (*n* = 71) or placebo (*n* = 67) | 36 | ↓Sister chromatid exchange ↓Pyridinoline, NF-B receptor activator ↑Alkaline phosphatase, IGF-I, and osteoprotegerin | [31] |

## 6. Synergistic Properties of Genistein in the Treatment of Breast Cancer

In addition to its solid anticancer activity alone, GNT possesses synergistic properties with many other anticancer drugs, helping it overcome chemopreventive resistance mechanisms in BC treatment. The synergistic activity of GNT can be carried by many anticancer drugs such as doxorubicin, trastuzumab, tamoxifen, trichostatin A, cisplatin, capsaicin, paclitaxel, and vincristine.

### 6.1. Synergistic Properties of Genistein with Doxorubicin in MCF-7/Adr Cells

Doxorubicin is an antibiotic that exhibits no inhibitory effects on Adriamycin-resistant BC cell lines. However, the combination of GNT at 30 μmol/L and doxorubicin has synergistic effects on MCF-7/Adr cells. GNT enhances the cytotoxic effects of doxorubicin and decreases the chemoresistance of MCF-7/Adr BC cells. In addition, GNT and doxorubicin synergistically induced apoptosis by decreasing expression of Her2/new mRNA and c-erbB2, resulting in cell cycle arrest in MCF-7/Adr BC cells in the G2/M phase [29]. Another study by Yang et al. reported that GNT (50 μM) with a combination of doxorubicin slightly induces apoptosis by destroying the plasma membrane of cells and increasing poly (ADP-ribose) polymerase cleavage in MDA-MB-231 and MCF-7 BC cells [92].

### 6.2. Synergistic Effect of Genistein with Trastuzumab

GNT and trastuzumab synergistically develop antitumor activity in BT-474 BC cells. C. Lattrich et al. stated that the combination of GNT (10 μmol/L) and trastuzumab (1/10 μg/mL) enhanced the growth-inhibitory effect and reduced viable cell numbers

by increasing the ER-β2 expression, which causes an antiestrogenic effect, leading to reduced cell proliferation in ER-α/β-positive and HER2-overexpressing BT-474 BC cells. Furthermore, both GNT and trastuzumab reduce cyclin A2 mRNA expression, c-Fos, HER2, and cyclin D1 expression, which suppresses the proliferation of BT-474 BC cells [152].

### 6.3. Synergistic Properties of Genistein with Tamoxifen

Tamoxifen is a well-established medicine for treating BC, although the development of gemcitabine resistance has hampered its efficacy. In this regard, GNT can improve the efficacy of tamoxifen. Y. Li et al. described that GNT enhanced the anticancer capacity of tamoxifen at the dose of 25 μM through the reactivation of ER-α and epigenetic pathway regulation, e.g., histone modification, resulting in the reduction in HDAC1 and DNMT1 expression both in vitro (ER-α-negative MDA-MB-231 BC cells) and in vivo, leading inhibited cell growth and cell viability [153]. On the other hand, Pons et al. concluded that 1 μM GNT and 10 μM tamoxifen decrease the ROS production in T47D and MCF-7 BC cells and upregulate the autophagic vacuole formation and PARP protein level and also reduce cell viability, resulting in autophagic cell death only in T47D BC cells [154]. Another early study reported that tamoxifen with GNT (1–10 μg/mL) impeded angiogenesis and cell mutation by downregulating ribosomal S6 kinases, tyrosine kinase, and cell cycle regulators [97]. GNT also shows a prohibitory effect with tamoxifen, which induces apoptosis by destroying the nuclear membrane [93] and arrests the cell cycle by decreasing the expression of HER2 in a dose-dependent manner [124].

### 6.4. Synergistic Effect of Genistein with Trichostatin A

GNT and trichostatin A act synergistically to inhibit PGR (progesterone receptors) expression, resulting in a significant change in cell growth in ER-positive and ER-negative BC cells. According to Li et al., the combination of GNT (25 μM) and trichostatin A (100 ng/mL) synergistically decreased ROS production through the underlying mechanism of increasing antioxidant enzymes, i.e., Mn-SOD (manganese-superoxide dismutase) and catalase, in MCF-7 and T47D BC cells. Furthermore, GNT and cisplatin also synergistically arrest the cell cycle at the S phase and cause a drop in the sub-G0/G1 phase, resulting in MCF-7 cells at 25–50 mol/L concentration [154].

### 6.5. Synergistic Effect of Genistein with Capsaicin

The combination of GNT and capsaicin exerted anti-inflammatory and anticarcinogenic effects in MCF-7 cells as well as in vivo in 48-week-old female Sprague–Dawley rats by modulating the mitogen-activated protein kinase (AMPK) and COX-2, as well as possibly other mitogen-activated protein kinases [30].

### 6.6. Synergistic Effect of Genistein with Paclitaxel and Vincristine

Paclitaxel and vincristine both are chemotherapy drugs. Together with GNT (100 μM), they can suppress cell growth and cell viability by inhibiting CDC2 and cyclin B1 kinase and inhibiting microtubule polymerization in human MDA-MB-231 and MCF-7 BC cell lines. Furthermore, cell death by inducing apoptosis via decreasing Bcl-2 phosphorylation without changing p21, p53, and Bax protein expression was also observed in combined treatment [103]. Table 3 summarizes the effects of phytoestrogens in combination with anticancer therapies that have been previously described.

**Table 3.** Summary of the described effects of phytoestrogens in combination with anticancer therapies.

| Treatment Option | Study Type | Effective Mechanism | Ref |
|---|---|---|---|
| Doxorubicin | In vitro (MCF-7/Adr) | ↓Chemoresistance of tumor cells<br>↑Cell cycle arrest $G_2$/M phase<br>↑Apoptosis<br>↓Her2/neu mRNA expression<br>↓c-erbB2 expression | [29] |
| | In vitro MCF-7 and MDA-MB-231 | ↑Destroyed the plasma membrane of cells<br>↑Apoptosis in most of the individual MCF-7 cells<br>↑Poly (ADP-ribose) polymerase cleavage | [92] |
| Trastuzumab | In vitro (SK-BR-3, BT-474, and MDA-MB-231) | ↓Cell proliferation<br>↑Cell cycle arrest<br>↓c-Fos<br>↓HER2 | [152] |
| Tamoxifen (TAM) | In vitro and vivo (MDA-MB-231) female immunodeficiency nude mice) | ↓Cell growth and cell viability<br>↓HDAC1<br>↓DNMT1 | [153] |
| | In vitro MCF-7 | ↑Destroyed the plasma membrane of cells<br>↑Nuclear membrane breakdown<br>↑*pS2* expression | [93] |
| | In vitro BT-474 | ↓Expression of HER2 and ER-$\alpha$<br>↓Expression of factor and EGFR | [124] |
| | In vitro dysplastic, malignant cells | ↓Cell growth, proliferation | [97] |
| Trichostatin A (TSA) | In vitro (MCF-7, MDA-MB-231, and MDA-MB-157) | ↑ER-$\alpha$ reactivation<br>↑Histone remodeling<br>↓HDACs, DNMTs, PGR expression | [153] |
| Cisplatin (CDDP) | In vitro (MCF-7 and T47D) | ↓ROS production<br>↑MnSOD, catalase activity<br>↑Arrest S and G0/G1 phase | [154] |
| Tamoxifen (TAM) | In vitro (MCF-7 and T47D) | ↓ROS, LC3-II/LC3-I ratio<br>↓Cell viability<br>↑PARP | [154] |
| Ornithine decarboxylase | In vitro MCF-7 MDA-MB-231 | ↑Putrescine, spermidine and spermine<br>↓12-Otetradecanoylphorbol-13-acetate (TPA)<br>↓Cyclooxygenase (COX) | [96] |
| Paclitaxel and vincristine | In vitro MCF-7 and MDA-MB-231 | ↓Bcl-2 phosphorylation<br>↓Cyclin B1 and CDC2 kinase<br>↓Cell viability<br>↓Microtubule polymerization | [103] |
| Lycopene | In vivo female Wistar rats | ↓Serum MDA<br>↓8-isoprostane and 8-OhdG levels<br>↓Bcl-2, caspase 3, and caspase 9<br>↓Expression of Bax | [155] |

## 7. Possible Strategies to Overcome Anticancer Drug Resistance by Genistein

Numerous mechanisms are responsible for BC drug resistance, such as membrane glycoproteins acting as efflux pumps, including P-glycoprotein (P-gp), multidrug resistance (MDR) protein, and BCRP, as well as enzymatic inactivation of the anticancer drug [156]. Phytochemical-based therapy can provide a reliable safety mechanism to prevent anticancer resistance. Inhibition of P-glycoprotein (P-gp) activities or conjunction of P-gp substrate with the anticancer drug leads to the increased accumulation of the anticancer drug within the cell, producing cell cytotoxicity. However, GNT does not directly affect P-gp function in a BC cell line but indirectly increases intracellular drug concentration, including doxorubicin. For instance, Castro and Altenberg stated that GNT decreased photo-affinity labeling of P-gp with [$^3$H] azidopine, a P-gp substrate, suggesting that GNT could block P-gp-mediated drug efflux by direct interaction with P-gp and inhibited rhodamine123 efflux in human MCF-7 BC cell lines [157]. Furthermore, intracellular doxorubicin accumulation was boosted by GNT therapy, leading to cell cycle arrest and apoptosis via inhibiting

HER2/neu rather than influencing P-gp function and MDR-1 expression in MCF-7/Adr cells [29]. GNT pretreatment with MDA-MB-231 and CB-17 scid/scid mice inactivated NF-κB and may contribute to increased growth inhibition and apoptosis induced by cisplatin docetaxel and doxorubicin in BC cells [158]. Targeting cyclooxygenase-2 (COX-2) can be a possible mechanism of overcoming drug resistance. There is a positive relationship between COX-2 and MDR1/P-gp. GNT significantly inhibited cyclooxygenase-2 activity [96], suggesting that GNT can inhibit MDR1/P-gp in BC, leading to improved anticancer drug efficacy [159]. In the case of BCRP, GNT and its glucuronide and sulfate conjugation are the substrates of BCRP established in in vitro cell culture models and in vivo pharmacokinetic studies [39,40]. This binding of GNT with BCRP indicates that GNT treatment increases anticancer drug concentration by decreasing efflux. However, there has been controversy with GNT and C member 1 (ABCC1) and ABCG2. Rigalli et al. reported that treatment of MCF-7 with GNT increases resistance to mitoxantrone and doxorubicin by increasing drug efflux [160].

## 8. Nano-Formulation of Genistein for Breast Cancer Treatment

GNT research for cancer treatment has been extended in recent years due to evidence of lower disease risk associated with its administration and a quest for pharmacological medicines that impact growth factor signaling pathways in cells. A significant drawback of GNT as a natural substance is its low water solubility. This may necessitate modifying its chemical structure to increase solubility and boost bioavailability [161].

However, the advancement of nanomedicines has the potential to overcome phytochemical limitations and allied health concerns, such as improved solubility, increased bioavailability, targeted treatment of tumor cells or tissues while avoiding healthy cell damage, and increased cell take-up. Nanomedicines could provide new avenues for circumventing these concerns. Additional advantages may include improved blood stability, multifunctional nanomedicine design, minimal interaction with synthetic medications, and improved anticancer activity [162]. Furthermore, multidrug resistance (MDR) is one of the most important variables contributing to the failure of phytochemical therapy in cancer. MDR can be circumvented using a new technique including nanocarriers and phytochemical delivery. Modifying the biophysical interaction between the nanomedicines and cancer cell membrane lipids can increase phytochemical delivery and overcome drug resistance. This is accomplished by improving the transport of phytochemicals to target tissues through surface modification of nanomedicines [163,164]. Currently, advancements in treatment efficiency through nanomedicines have received much attention because of the increased delivery of phytochemicals to tumors and cancer cells. Numerous highly successful nanomedicines have been employed to enhance phytochemicals' physicochemical qualities and anticancer activity [165]. BC treatment with doxorubicin and GNT is improved by using multifunctional hybrid nano-constructs that enable intracellular localization of the drugs [166]. A research study by Jimmy Pham and his colleagues demonstrated that mitochondriotropic nano-emulsified genistein-loaded vehicles showed more effective potential against hepatic and colon carcinomas than the control drugs [167]. In one study, cervical cancers were treated with bioflavonoid genistein-loaded chitosan nanoparticles targeted to the folate receptor, which had a significant anticancer effect. The naturally derived chitosan nanoparticles exhibited potent biodegradability and biocompatibility when coated with the GNT [168]. Additionally, genistein-loaded biodegradable TPGS-b-PCL nanoparticles possessed enhanced therapeutic effects in cervical cancer cells [169]. Moreover, the nanoformulation of GNT promoted selective apoptosis in the cell line of oral squamous cancer by suppressing the expression of a 3PK-EZH2 signaling pathway [170].

## 9. Concluding Remarks and Future Directions

The evidence provided from available scientific literature (in vitro and in vivo) detailed in this review offers a comprehensive summary of the anticancer activities of GNT. Overall, information from our study would help in the identification of the mechanisms

of GNT against BC pathogenesis that will aid in drug development in anti-BC therapy. Mechanisms of GNT are related to multiple molecular pathways, including regulating miRNA, several proteins such as apoptosis, transcription factor and tumor suppressor-related proteins, enzymes including kinase, several growth factors, receptors, and other numerous targets (Figure 2). Successful conventional treatment of BC is limited due to rising resistance to some chemotherapeutic drugs, but GNT may bring therapeutic advantages by sensitizing multidrug-resistant BC cells and mediating some synergistic effects with conventional anticancer drugs. Therefore, using GNT as a regular food supplement may help in the near future to treat BC. However, we still need further research, including clinical trials, regarding drug interactions, accurate pharmacokinetics, accurate therapeutics doses, routes of administration, and established nanoformulation of GNT. The successful performance of all approaches will make GNT a novel candidate for drug development against BC.

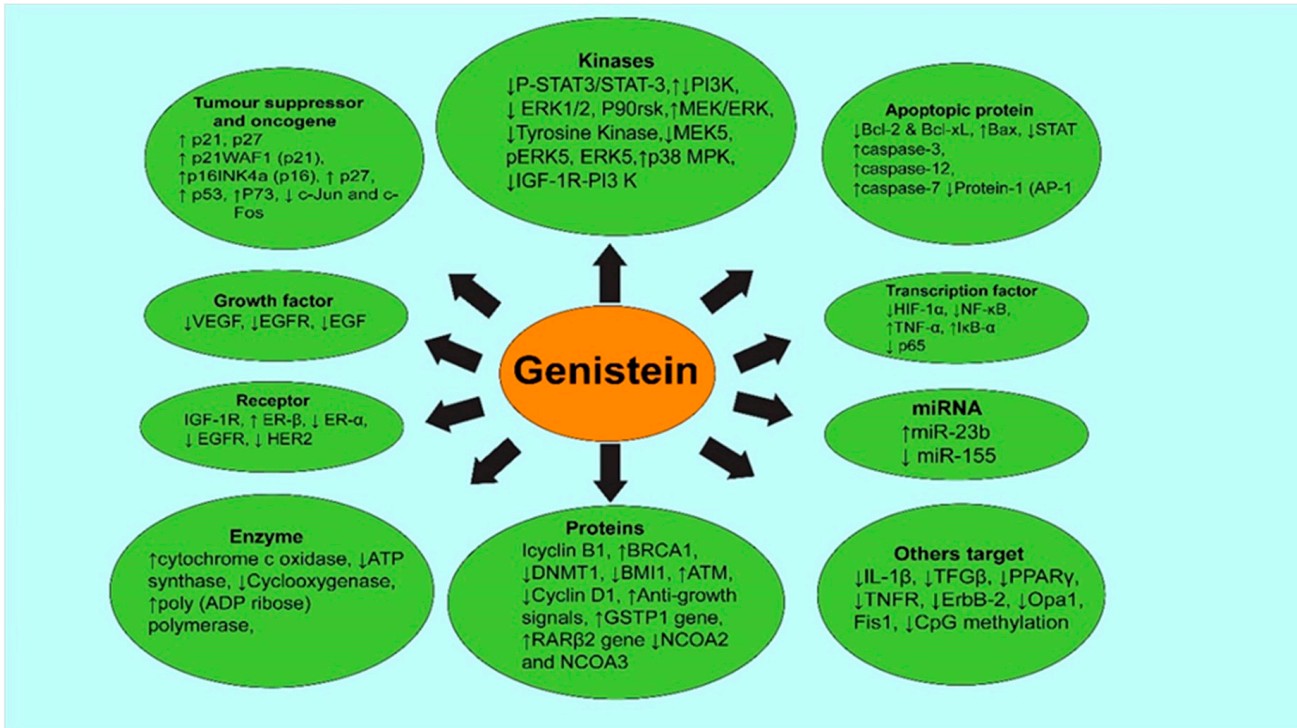

**Figure 2.** Overview of molecular targets influenced by GNT in breast cancer.

**Author Contributions:** M.S. contributed to the conceptualization and study design; M.S., P.B., M.A.A., M.A.H., H.S., D.D., S.A., M.S.K., P.P., A.S. and M.N.I. equally participated in data collection, writing, and draft preparation; M.S., M.A.H. and H.S. contributed to table and figure generation; M.A.R. and A.A.M. contributed to review and editing; M.A.R., A.A.M. and B.K. contributed to visualization and supervision; B.K. contributed to funding. All authors have read and agreed to the published version of the manuscript.

**Funding:** This research was supported by the Basic Science Research Program through the National Research Foundation of Korea (NRF) funded by the Ministry of Education (NRF-2020R1I1A2066868); the National Research Foundation of Korea (NRF) through a grant funded by the Korean government (MSIT) (no. 2020R1A5A2019413); a grant of the Korea Health Technology R&D Project through the Korea Health Industry Development Institute (KHIDI), funded by the Ministry of Health and Welfare, Republic of Korea (grant number HF20C0116); and a grant of the Korea Health Technology R&D Project through the Korea Health Industry Development Institute (KHIDI), funded by the Ministry of Health and Welfare, Republic of Korea (grant number HF20C0038).

**Institutional Review Board Statement:** Not applicable.

**Informed Consent Statement:** Not applicable.

**Data Availability Statement:** The datasets used and/or analyzed during this study are available from the corresponding authors upon request.

**Conflicts of Interest:** The authors declare no conflict of interest.

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
