# Peer review of "Genistein, a Potential Phytochemical against Breast Cancer Treatment-Insight into the Molecular Mechanisms"

_processes, doi:10.3390/pr10020415_

Round 1

Reviewer 1 Report

Thanks for considering my edits to this manuscript. Still, there are a few things to be addressed.

1. Protein labels in Figure 1 are so small. That figure also looks crowded and confusing. This figure needs to be re-organized.

Author Response

Thanks for considering my edits to this manuscript. Still, there are a few things to be addressed. Protein labels in Figure 1 are so small. That figure also looks crowded and confusing. This figure needs to be re-organized.

Response>>

  • The full manuscript was gone through and added all requirements
  • Figure 1 was reorganized by addressing all possible proteins were available in the manuscript. Now the visualization of the figure is clearer than the previous one.

Reviewer 2 Report

The major flaws encountered in the previous submission have been addressed in this resubmitted manuscript. 

Author Response

The major flaws encountered in the previous submission have been addressed in this resubmitted manuscript. 

Response>> Thank you to the reviewer for comments.

Reviewer 3 Report

This is a valuable work for the scientific community. It is highly recommended that you should employ the services of a professional editor to assist with editing this paper. I noticed significant grammatical errors and typographical mistakes. This makes part of the work difficult to understand. In your introduction line 114 you stated that "there is no clear evidence that consumption of large amount of GNT is harmful" however, in line 85 you shared that "..GNT promotes malignant growth" These two statements are contradictory and it is important to review this Overall, good work.

Author Response

This is a valuable work for the scientific community. It is highly recommended that you should employ the services of a professional editor to assist with editing this paper. I noticed significant grammatical errors and typographical mistakes. This makes part of the work difficult to understand. In your introduction line 114 you stated that "there is no clear evidence that consumption of large amount of GNT is harmful" however, in line 85 you shared that "..GNT promotes malignant growth" These two statements are contradictory and it is important to review this. Overall, good work

Response>>

  • All grammatical errors and typographical mistakes were cheeked properly and corrected throughout the manuscript. All of the changed portions were marked as blue color.
  • Line 114 you stated that "there is no clear evidence that consumption of large amount of GNT is harmful” was modified into “Although there is limited evidence that consuming large amounts of GNT in the diet causes a deleterious effect in humans, but the toxicity of GNT on fertility and fetal development has been extensively studied in recent years”
  • “GNT promotes malignant growth”. Yes. Some researchers claim some contradictory activities of GNT and promotion of malignant growth is one of them. But this happens in rare cases. So we have proved the beneficial effect of GNT rather than the contradictory effect.

This manuscript is a resubmission of an earlier submission. The following is a list of the peer review reports and author responses from that submission.

Round 1

Reviewer 1 Report

In this paper, the authors reviewed the molecular pharmacology of the compound called Genistein in breast cancer.  In the abstract and introduction, there are lots of critical misconceptions regarding breast cancer literature and statistics. Therefore, I do want to reject this paper due to ethical reasons.

Reviewer 2 Report

Dear authors of manuscript #processes-1393290, which is supposed to be a review article titled “Genistein Mediated Molecular Pharmacology, Cell-Specific Anti-Breast Cancer Mechanism with Synergistic Effect and in silico Safety Measurement”. I appreciate your attempt to review the phytochemical genistein in breast cancer. Yes, genistein is an interesting natural product that was reviewed for interesting bioactivities in several recent well-written literatures. But, in my humble opinion, the current manuscript is full of mistakes, imprecise and/or inaccurate information, and sometimes exaggerations and inappropriate language. In general, it is not suitable for publication. I suggest withdrawal or rejection after that you may rewrite again and resubmit it. If rewrite it, please consider that subjectivity, freeness of errors, accurateness and preciseness are basics in writing scientific article. In addition, a narrative review article should highlight key points in the topic and provide authors’ message or viewpoint regarding the addressed topic rather than just presenting summary of other researchers published work. I regret to add that I think that presented in silico ADMET evaluation is futile exercise. I cannot provide list of all issues found in the manuscript because they are so many and huge in number and exists almost in every sentence of the manuscript. I just provide very limited examples from the abstract, introduction and first page after the introduction:

In the abstract:

  1. The first sentence “Breast cancer (BC) is the most common type of cancer in both men and women” presents a false claim probably written in attempt to exaggerate the burden of breast cancer!!! Of course, breast cancer is NOT the most common cancer in men but the most common cancers in men are prostate cancer, lung cancer, colorectal cancer, bladder cancer, melanoma.

  1. The second sentence “Natural compounds that can modulate the oncogenic process can be considered a significant anti-cancer agent for treating BC.” is inaccurate! Can any natural compound modulating any oncogenic process show significant anti-breast cancer activity? I doubt!

  1. The third sentence “These natural compounds are more effective than synthetic drugs, which have profound side effects on the normal cell and resistance to cancer cells.” is wrong. Some synthetic drugs are highly effective and, furthermore, natural products can affect normal cells and might be subject to evolvement of resistance.

…. and so on in the remainder of the abstract!

In the introduction:

  1. The first sentence “…. a major public health concern among women in health science” is not appropriate language. It implies that only women working in health science are interested in breast cancer while men working in health science are not interested!

  1. The second sentence “BC is …… and is the world's second leading cause of mortality”. This is wrong claim. Not breast cancer but all cancers collectively are the second cause of death after cardiovascular diseases.

  1. Next, “There are currently no established treatments are available” is another false claim. There are available treatments but might have limitations and, therefore, better treatments are required.

…. and so on in the remainder of the introduction!

In the first page after the introduction:

  1. In the first sentence (line 107), “it inhibits the BC cell proliferation at high concentrations”. What is the range of this high concentrations?

  1. In the second sentence (lines 107 and 108), “Estrogen receptor-mediated growth of BC cells by Gen is concentration-dependent”!!! Growth or inhibition of growth? Is this for all breast cancers or found for a specific cell line?!

  1. “Genistein has a similar structure both of these types of receptors but binds to estrogen receptor-β (ER-β) with higher affinity compared to estrogen receptor-α (ER-α)”!!! This is a broken language that requires the reader to try guessing what authors wish to say!

  1. Line 117, “(full elaboration add here)”!!!! Are you asking the reader to add full elaboration by himself? How can you request the reader to do this?!

….. and so on in all other sections of the manuscript. There are huge number of errors and unacceptable mistakes, false or inaccurate claims.                

Reviewer 3 Report

The work of Sohel et.al entitled Genistein Mediated Molecular Pharmacology, Cell-Specific Anti-Breast 3 Cancer Mechanism with Synergistic Effect and in silico Safety Measurement describest in great details the effect of genistein on breast cancer cells. The authors emphasize that compounds of natural origin can be used not only in the prevention but also in the treatment of neoplastic diseases. The antiproliferative, cytotoxic and pro-apoptotic effects of genistein are exemplified by several molecularly differentiated breast cancer cell lines. The possibility of using genistein together with routinely used, synthetic chemotherapeutic agents is presented in a very interesting way. Particularly noteworthy is the chapter devoted to clinical studies on the effects of genistein - I believe that it could be described in more detail. Figures included in the work are very aesthetic and accurate, they help to better understand the molecular mechanism of action of the tested compound. The tables, in turn, systematize the knowledge contained in the text. The authors cite over 100 works, most of them are literature from the last 10 years, which is why I believe that the topic of the work has been thoroughly discussed, and despite the fact that the beneficial effect of genistein on oncological patients has long been confirmed, the work brings a lot of valuable content